# Heterogeneous Chitosan@copper Catalyzed Selective C(sp³)–H Sulfonylation of Ketone Hydrazones with Sodium Sulfinates: Direct Access to β-Ketosulfones

Jun Qiao [1,2], Kai Zheng [2], Zhiwei Lin [2], Huiye Jin [2], Wenbo Yu [3], Chao Shen [2,*], Aiquan Jia [1] and Qianfeng Zhang [1,*]

1   School of Materials Science and Engineering, Institute of Molecular Engineering and Applied Chemistry, Anhui University of Technology, Ma'anshan 243002, China
2   College of Biology and Environmental Engineering, Key Laboratory of Pollution Exposure and Health Intervention of Zhejiang Province, Zhejiang Shuren University, Hangzhou 310015, China
3   Linjiang College, Hangzhou Vocational and Technical College, Hangzhou 310018, China
*   Correspondence: shenchaozju@163.com (C.S.); zhangqf@ahut.edu.cn (Q.Z.)

**Abstract:** The exploration of inexpensive and stable heterogeneous catalysts for C–S coupling reactions remains a challenging issue. Herein, we successfully prepared a new biomass-derived copper catalyst and applied it to the selective C(sp³)–H-directed sulfonylation of ketone hydrazones with commercial sodium sulfinates. The catalyst was characterized using different spectroscopic and microscopic techniques. Importantly, the prepared biomass-supported Cu catalyst catalyzed the C–S coupling reaction with considerably high activity. A variety of aryl and alkyl sulfinates were converted to the corresponding sulfones with good yields. In particular, the heterogeneous catalyst could be recovered easily after the synthesis and consecutively used at least five times with no appreciable decrease in the catalytic activity. Lastly, the copper catalyst is expected to have further applications in organic reactions catalyzed by Cu/CuO$_2$.

**Keywords:** biomass-derived; copper catalysts; heterogeneous; C–S coupling

## 1. Introduction

Organosulfur compounds play an essential role in organic chemistry, as well as in medicinal chemistry [1–5]. Hence, particular interest has been focused on the synthesis of organosulfur compounds due to their potential to serve as novel pharmaceutical, agricultural, and chemical agents [6]. Among the organosulfur compounds, β-ketosulfones have acquired increased relevance in organic and medicinal chemistry in the past few years [7–10]. This is not only due to their important application in organic synthesis but also to their exhibiting a spectrum of biological activity (Figure 1) [11–15]. Recently, sulfonylation has drawn significant intense attention as one of the promising synthetic methods for the preparation of β-ketosulfones and their analogs [16–19]. The general approach to the synthesis of β-ketosulfones is the nucleophilic substitution reaction of sodium sulfonates with α-haloketone [20–22]. One representative method for the preparation of β-ketosulfones involves Cu-catalyzed direct C–S cross-coupling of various oxime acetates with sodium sulfinates to construct a C(sp³)–S bond reaction [23–36]. Alternatively, various β-ketosulfones can be prepared through the Cu-catalyzed direct C–S coupling reaction of substituted alkenes with various sulfonylhydrazides under an oxygen atmosphere. Recently, Jiang's group reported an interesting two-step synthetic method that involved the oxidative coupling of sodium sulfinates and oxime acetates using copper (Scheme 1a) [37]. Drawing inspiration from this work, Xia et al. employed oxidative coupling of oxime acetates with sodium sulfinates under heterogeneous catalysis (MCM-41-Py-Cu(OAc)$_2$) (Scheme 1b) [38]. Similarly, Phan's group developed a metal-organic framework (MOF)-based Cu$_2$(OBA)$_2$(BPY) catalyst that was used as a recyclable heterogeneous

catalyst for oxosulfonylation (Scheme 1c) [39]. In 2021, Liu's group further synthesized β-ketosulfones from hydrazones and sodium sulfinates by synergistic copper/silver catalysis (Scheme 1d) [40]. However, despite notable advancements obtained in this area [41–44], the development of a recyclable and facile protocol for the construction of biologically active β-ketosulfones remains highly desirable.

**Figure 1.** Examples of biologically active β-ketosulfones.

Developing an advanced heterogeneous catalyst is essential for realizing high-performance C–S coupling reactions. Chitosan (CS) is an abundant, sustainable, and biodegradable green material with numerous functionalities [45]. It is mostly produced from crab or shrimp shell-derived chitin and bears a lot of primary amino, acetylamino, and hydroxy groups available for binding varied metal ions [46]. Therefore, chitosan has become important biomass support and has attracted significant interest in heterogeneous catalysis [47,48].

In addition, a variety of biomass-derived catalysts were synthesized from chitosan and metal salts by pyrolysis at various temperatures [49]. As one of the requirements of green chemistry, non-precious transition metals catalysts based on earth-abundant biomass are also applied in organic synthesis. For example, Beller and coworkers developed novel heterogeneous chitosan-based cobalt catalysts that were applied in the catalytic hydrogenation of nitroarenes and hydrodehalogenation of aryl halides with outstanding catalytic performance [50]. Our group developed a chitosan-based Cu(OAc)$_2$ catalyst and applied it in the trifluoromethylation of quinoline compounds with the inexpensive CF$_3$SO$_2$Na through direct C–H activation [51]. However, this approach encounters several challenging issues, such as activity, stability, and selectivity for various organic reactions.

In continuation of our research on heterogeneous biomass-derived catalysts [52,53], we have paid great attention to the preparation of pyrolyzed chitosan-based materials. Herein, a straightforward concept was presented for the synthesis of novel eco-friendly and reusable heterogeneous catalysts based on the pyrolysis of complexes of copper salts and chitosan. The as-prepared catalysts (Cu(OAc)$_2$/chitosan pyrolyzed at 400 °C) were used for synthesizing various β-ketosulfones via direct C–S coupling of ketone hydrazone with sodium sulfinates. The advantages of the present protocol include the usage of heterogeneous catalysts, mild reaction conditions, and good yields of the products.

**Previous Work**

**Scheme 1.** Different synthetic approaches to β-ketosulfones. (**a**)—Recation of Sodium Sulfinates with Oxime Acetates, Ref. [37]; (**b**)—Recation of Sodium Sulfinates with Oxime Acetates, Ref. [38]; (**c**)—Recation of Sodium Sulfinates with Oxime Acetates, Ref. [39]; (**d**)—Recation of Sodium Sulfinates with Ketone Hydrazones, Ref. [40].

## 2. Results and Discussion

The precursors of our novel catalysts were prepared by the impregnation method. Firstly, the commercial chitosan was added to an aqueous solution of $Cu(OAc)_2$ to obtain a suspension that was stirred for 3 h at 50 °C. After cooling the reaction mixture to room temperature, the copper@CS solid was obtained through the separation of the water by reducing the pressure and dried under vacuum at 60 °C overnight. Then, the dried samples were transferred into a porcelain boat and placed in an oven. The oven was flushed with nitrogen, followed by pyrolysis at 300, 400, and 500 °C. The obtained chitosan-based catalysts were named $Cu_xO_y$@CS-T, where T denotes the pyrolysis temperature (Scheme 2). Finally, the as-prepared catalysts were stored in screw-capped vials without any special protection from the air at room temperature. The most active catalyst obtained by pyrolysis at 400 °C was fully characterized by various methods. Fourier transform infrared (FTIR) spectroscopy is a significant strategy to research catalyst structure. The FTIR spectrum of the $Cu_xO_y$@CS-T catalyst exhibits characteristic bands that were similar to the chitosan structure (Figure S1). The peak at about 3449 cm$^{-1}$ corresponded to the associated -OH groups and N–H stretching [54]. The weak peak that appears at approximately 2973 cm$^{-1}$ was assigned to C–H stretching [55]. The band at 1658 cm$^{-1}$

was attributed to the amide vibration of the remaining acetyl groups of chitosan [54,55]. The characteristic band at 1031 cm$^{-1}$ was attributed to C–O stretching vibrations [55]. Typical characteristic peaks such as –OH and –NH$_2$ were found in the characterization of the carbon materials by FTIR, which demonstrated that the carbon materials have excellent adsorption capability for metals. Thermogravimetry (TG) obtained during the heat treatment of the Cu(OAc)$_2$-Chitosan complex under a nitrogen atmosphere is shown in Figure S2. There was a rapid mass loss in the temperature range from 250 °C to 400 °C, which could be ascribed to the degradation of chitosan polymer and acetate counteranion. The catalyst material yield for the pyrolysis at 400 °C was 54.8 wt%. Scanning electron and transmission electron micrographs obtained from the prepared Cu$_x$O$_y$@CS-400 catalyst are shown in Figure 2. The results indicate that the Cu was evenly distributed throughout the catalyst, and no agglomerated particles could be observed. It was also evidenced that the copper nanoparticles were successfully immobilized with the pyrolyzed chitosan (Figure 2a–d). The element mapping performed with the prepared Cu$_x$O$_y$@CS-400 catalyst showed the homogeneous presence of carbon, nitrogen, and oxygen together with copper elements(Figure 2e–j).

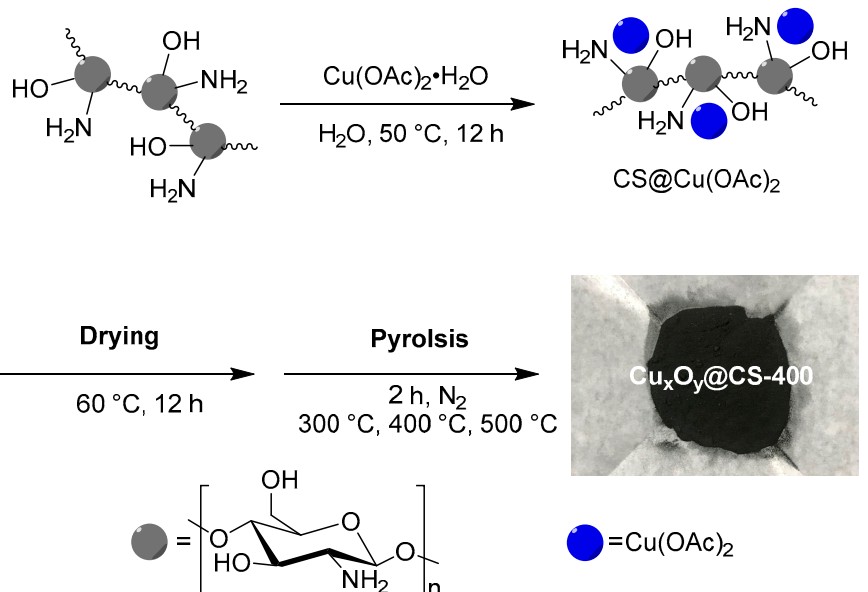

**Scheme 2.** Synthesis of copper@CS catalysts.

The surface chemical composition of the catalyst was characterized by X-ray photoelectron spectroscopy (XPS), and peaks corresponding to C 1s, N 1s, O 1s, and Cu 2p were observed in the XPS survey spectrum, which confirmed the elemental composition of the catalyst (Figure 3a). Figure 3b shows the binding energy at 932.4 eV and 952.2 eV (Cu 2p3/2 and 2p1/2) with the appearance of satellite peaks, which strongly demonstrated that the Cu or Cu$_2$O state was present. The Cu 2p spectrum also showed the presence of four distinct peaks at 934.3, 943.0, 954.3, and 962.5.3 eV, which were attributed to CuO. As shown in the O 1s spectrum (Figure 3c), the binding energy peak for Cu$_2$O was observed at 531.0 eV. Moreover, the binding energy peak at 533.3 eV suggested the presence of surface –OH in the catalyst [56,57]. The chemical states of copper in the catalyst were further investigated by X-ray-induced Cu LMM Auger spectroscopy (Figure 3d).

There was an Auger peak in the XPS spectra of the prepared catalyst at 570.2 eV. From this, we can conclude that Cu$_2$O was present on the surface of the catalyst [58–60]. Additionally, the catalyst was characterized by XRD (Figure 3e). As can be seen, the Cu$_x$O$_y$@CS-400 catalyst mainly showed sharp diffraction peaks of Cu and relatively weak peaks of Cu$_2$O. The diffraction peaks at 43.2° and 50.4° are assigned to the (111) and (200) reflections of Cu (JCPDS NO.04-0836). The peaks observed at 29.5°, 36.4°, 42.3°, and

61.3° correspond to the (110), (111), (200), and (220) reflections of $Cu_2O$ (JCPDS NO.05-0667) [60,61], In addition, a broad diffraction peak centered at 24.2°, which can be assigned to (002) reflections of amorphous carbon phase [62]. These analysis results verified that the catalyst structure contained Cu and CuO. To sum up, the XPS and XRD further confirmed the existence of Cu, $Cu_2O$, and CuO in the prepared catalyst.

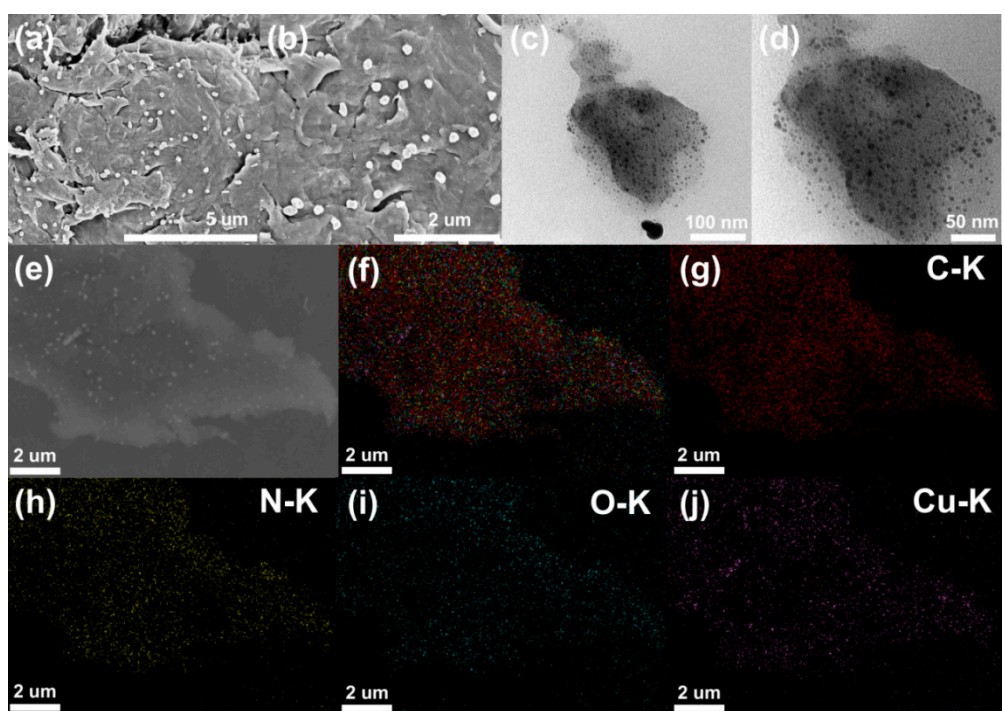

**Figure 2.** SEM images of $Cu_xO_y@CS$-400 (**a**,**b**). TEM images of $Cu_xO_y@CS$-400 (**c**,**d**). Elemental mapping of $Cu_xO_y@CS$-400 (**e**−**j**).

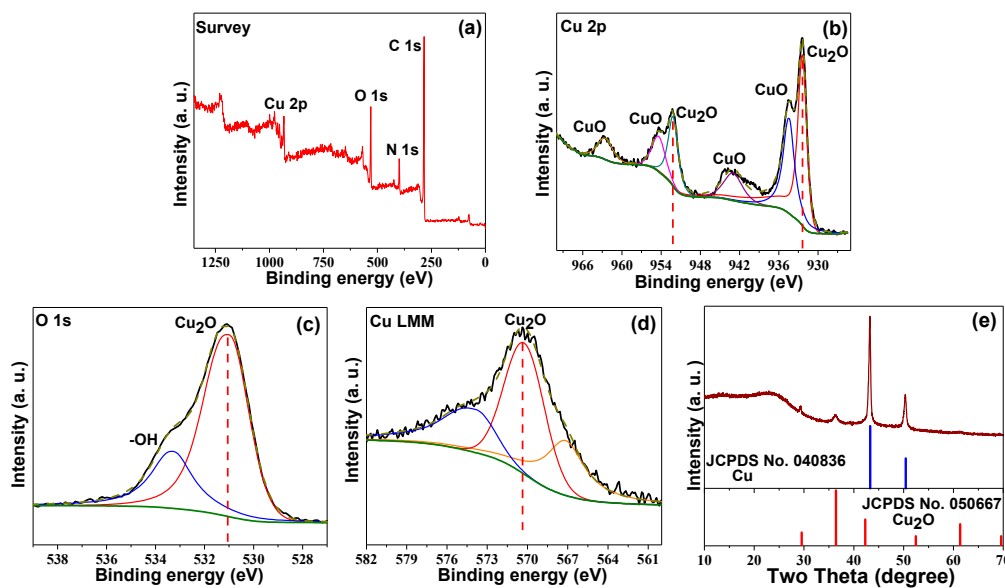

**Figure 3.** XPS spectra of $Cu_xO_y@CS$-400: (**a**) Survey spectrum. (**b**) Cu 2p. (**c**) O 1s. (**d**) Cu LMM. (**e**) XRD spectra of $Cu_xO_y@CS$-400.

With the catalysts in hand, we chose the selective C–S coupling of acetophenone hydrazone **1a** with sodium sulfinate **2a** as the model reaction. In a preliminary test, the

desired coupling product **3a** was obtained in 56% yield with 30 mg of Cu$_x$O$_y$@CS-400, 20 mol% of Ag$_2$CO$_3$, and 2.0 equivalents of K$_2$S$_2$O$_8$ as the oxidant in acetone. Encouraged by this result, the Cu$_x$O$_y$@CS-300 and Cu$_x$O$_y$@CS-500 catalysts were applied for the sulfonylation reaction, respectively. The Cu$_x$O$_y$@CS-400 was found to be the most efficient and afforded **3a** in 56% yield, which indicates that the reaction efficiency followed the order: Cu$_x$O$_y$@CS-400 > Cu$_x$O$_y$@CS-300 > Cu$_x$O$_y$@CS-500 (Table 1, entries 1–3). Subsequently, other Cu catalysts, such as Cu powder, Cu$_2$O, and heterogeneous catalysts prepared by our group (Table S1) were also investigated, and the results showed the poor formation of the target product (Table 1, entries 4–6). It was demonstrated that the active substance Cu$_2$O has a direct effect on the activity of the catalyst. Due to the pyrolysis product of copper(II) acetate being greatly affected by operating conditions, the Cu$_x$O$_y$@CS-400 catalyst contains a more active substance Cu$_2$O than other pyrolysis catalysts [63]. The sulfonylation reaction didn't proceed in the absence of a Cu catalyst (Table 1, entry 7). To increase the formation of product **3a**, reaction conditions, including types of oxidants, solvents, and temperatures, were screened. To our surprise, the yield of the desired product **3a** could be significantly enhanced to 81% when the sulfonylation reaction was performed using K$_2$S$_2$O$_8$ as the oxidant (Table 1, entry 8), and the yield was only 41% when (NH$_4$)$_2$S$_2$O$_8$ was replaced with K$_2$S$_2$O$_8$ (Table 1, entry 9). Subsequently, several solvents were tested; when using MeCN, H$_2$O, and toluene as the solvents, the yields of product **3a** were 23, 0, and 35%, respectively (Table 1, entries 10–12). Changing the loading of the catalyst or reaction temperature could not increase the reaction efficiency (Table 1, entry 13–15). Therefore, the optimized results for the copper-catalyzed direct C–S coupling reaction were achieved with acetophenone hydrazone **1a**, sodium benzenesulfinate **2a**, Cu$_x$O$_y$@CS-400, silver carbonate, potassium persulfate was carried out in acetone at 80 °C under air for 5 h.

**Table 1.** Control experiments [a].

| Entry | Catalyst (mol %) | Oxidant | Solvent | Yield (%) [b] |
|---|---|---|---|---|
| 1 | Cu$_x$O$_y$@CS-300 | Na$_2$S$_2$O$_8$ | acetone | 21 |
| 2 | Cu$_x$O$_y$@CS-400 | Na$_2$S$_2$O$_8$ | acetone | 56 |
| 3 | Cu$_x$O$_y$@CS-500 | Na$_2$S$_2$O$_8$ | acetone | trace |
| 4 | Cu | Na$_2$S$_2$O$_8$ | acetone | trace |
| 5 | Cu$_2$O | Na$_2$S$_2$O$_8$ | acetone | 55 |
| 6 | CuO | Na$_2$S$_2$O$_8$ | acetone | trace |
| 7 | - | Na$_2$S$_2$O$_8$ | acetone | 0 |
| 8 | Cu$_x$O$_y$@CS-400 | K$_2$S$_2$O$_8$ | acetone | 81 |
| 9 | Cu$_x$O$_y$@CS-400 | (NH$_4$)$_2$S$_2$O$_8$ | acetone | 41 |
| 10 | Cu$_x$O$_y$@CS-400 | K$_2$S$_2$O$_8$ | MeCN | 23 |
| 11 | Cu$_x$O$_y$@CS-400 | K$_2$S$_2$O$_8$ | H$_2$O | 0 |
| 12 | Cu$_x$O$_y$@CS-400 | K$_2$S$_2$O$_8$ | toluene | 35 |
| 13 | Cu$_x$O$_y$@CS-400 | K$_2$S$_2$O$_8$ | acetone | 82 [c] |
| 14 | Cu$_x$O$_y$@CS-400 | K$_2$S$_2$O$_8$ | acetone | 57 [d] |
| 15 | Cu$_x$O$_y$@CS-400 | K$_2$S$_2$O$_8$ | acetone | 80 [e] |

[a] Reaction conditions: **1a** (0.2 mmol), **2a** (0.4 mmol), catalyst (30 mg), silver carbonate (20 mol%), oxidant (2.0 equiv), solvent (1.5 mL), stirred at 80 °C, under air, 5 h. [b] Isolated yields. [c] 40 mg of catalyst was used. [d] 60 °C. [e] 100 °C.

Having successfully achieved the sulfonylation of acetophenone hydrazone **1a**, the scope of the ketone hydrazone was explored (Figure 4). The ortho-substituted substrate ketone hydrazone **1b** reacted well with sodium benzenesulfinate **2a** to form the C–S cou-

pling product **3b** in 70% yield, indicating that the reaction was hardly subjected to steric hindrance. Notably, when the ketone hydrazone **1** bearing substituents at a meta position (Cl and CH$_3$), the products **3c** and **3d** were furnished in good yields of 77% and 81%. Ketone hydrazone with various para- or meta-substituents that were either electron-donating groups (CH$_3$) or electron-withdrawing groups (NO$_2$, F, Cl, Br) showed satisfactory compatibility, and the β-ketosulfones **3e–3j** were produced in this reaction in 67–88% yields. Meanwhile, propiophenone hydrazones and heterocyclic ketone also could undergo the reaction smoothly, providing the corresponding products in 83% (**3k**) and 74% (**3l**) yield, respectively.

**Figure 4.** The scope of the ketone hydrazone **1** and sodium sulfinates **2** [a,b]. [a] Reaction conditions: **1** (0.2 mmol), **2** (0.4mmol), catalyst (30 mg), Ag$_2$CO$_3$ (20 mol%), K$_2$S$_2$O$_8$ (2.0 equiv), acetone (1.5 mL), stirred at 80 °C, under air, 5 h. [b] Isolated yields.

Some sodium sulfinates were also explored with **1a** as a C–S coupling partner. It was found that sodium sulfinates regardless of whether they had electron-withdrawing or -donating groups gave the coupling products with satisfying yields. The sulfonylation reaction of a variety of para-substituted sodium benzenesulfinates (CH$_3$, CF$_3$, OMe, Cl, or Br) also reacted smoothly with **1a** to furnish coupling products **3r–3v** in good yields. Furthermore, the meta- and ortho-substituted sodium benzenesulfinates have been transformed into the products **3m–3q** with yields of 65% to 81%. Pleasingly, the aliphatic sodium sulfinates showed good reactivity to form the corresponding β-ketosulfones products in 63% (**3w**) and 67% (**3x**) yield, respectively.

To demonstrate the potential applications of this strategy, the derivatization and the gram-scale preparation of β-ketosulfones were examined (Scheme 3). The desired products **5a** and **5b** could be easily obtained with good yields. Furthermore, the bioactive molecule **3y** was synthesized on a gram scale with an 80% yield. These results clearly show the practicality of the strategy in organic synthesis.

**Scheme 3.** Derivatization and gram-scale experiments and corresponding reference.

Both stability and reusability are important parameters for transition-metal-catalyzed reactions. Further experiments were performed to verify the catalyst recyclability using the reaction of acetophenone hydrazone **1a** with sodium benzenesulfinate **2a** as a model system. As depicted in Figure 5, after five successive cycles, affording the desired product **3a** in 78% yield. The catalytic performance is only slightly affected. In addition, a hot filtration experiment was performed to determine whether the transformation was derived from the catalyst or a leached copper species. The sulfonylation reaction between acetophenone hydrazone **1a** and sodium benzenesulfinate **2a** under the optimized conditions was conducted. After 2 h, the $Cu_xO_y$@CS-400 catalyst was hot-filtered and the catalyst-free solution was performed continuously at 80 °C under air for 12 h. However, the yield of the desired product **3a** did not increase. No measurable leaching of copper from the catalyst by ICP-AES analysis after the first and fifth runs was observed.

To gain more insight into the mechanism of the sulfonylation process, some control experiments were designed and conducted (Scheme 4). Firstly, β-ketosulfones could not be obtained without the assistance of hydrazone, which revealed the importance of hydrazone in this transformation. The addition of the radical scavenger 2,2,6,6-tetramethyl piperidine-1-oxyl (TEMPO) into our standard reaction did not lead to the formation of product **3a**, which indicated that the formation of product **3a** involved a radical pathway. Moreover, when adding the radical trapping reagent 1,1-diphenylethylene (DPE), the resulting sulfonyl radical was captured. These mechanism experiments indicated that the radical mechanism proposed by us was reasonable. Conversely (Scheme S1), replacing

air with $N_2$ did not affect the formation of **3a**, indicating that $O_2$ from the air was not a key oxidant participating in the reaction. Thereafter, the reaction was performed in the presence of 5 equiv of $H_2O^{18}$, and a β-ketosulfone containing $O^{18}$ was detected, which indicates that the oxygen on the carbonyl should come from $H_2O$ (Scheme 4). In addition, the intermediate product **D** was formed in the presence of dry acetone (Scheme 4). These results showed that the formation of the β-ketosulfone was water-assisted.

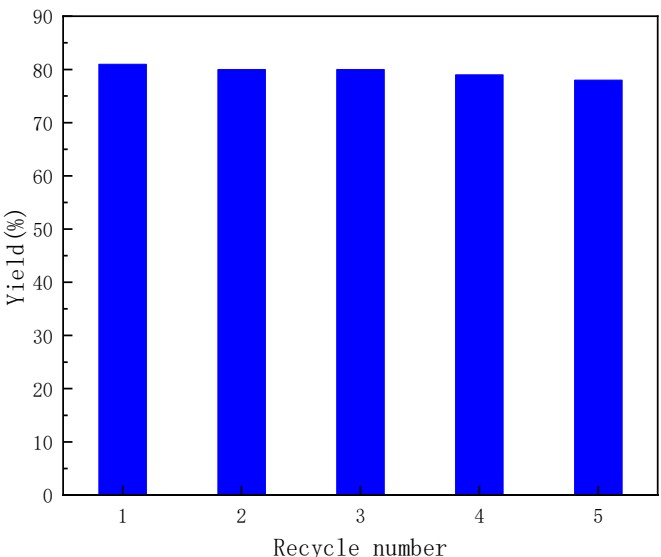

**Figure 5.** Catalyst recycling for the C–S coupling reaction.

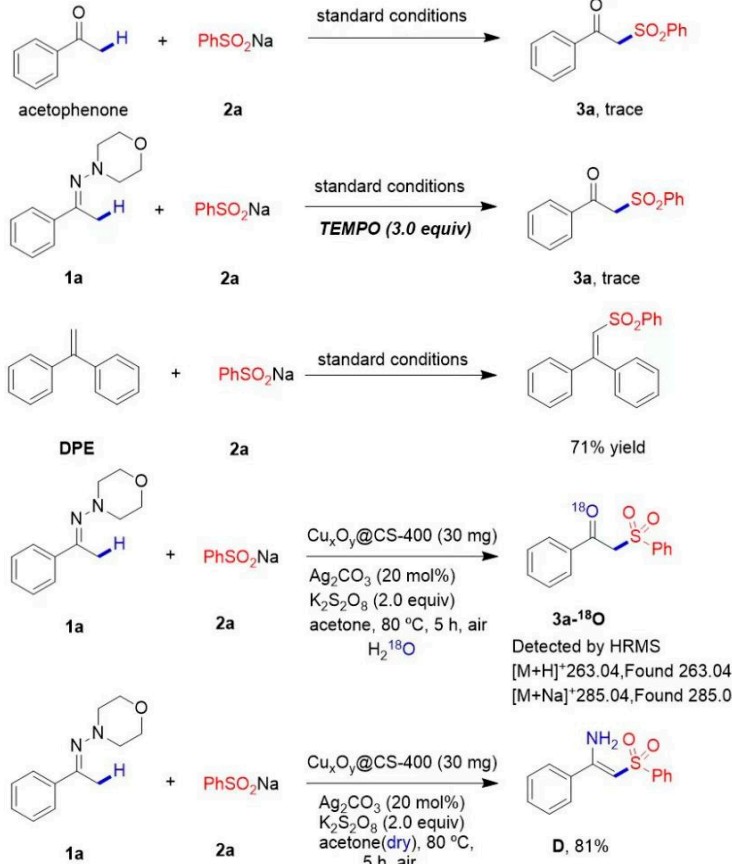

**Scheme 4.** Mechanism study.

A plausible mechanism (Scheme 5) for the C–S coupling reaction was proposed based on the results of the above control studies and the literature [37,40]. Firstly, a copper imine intermediate **A** was formed from the acetophenone hydrazone **1a,** while $Cu^I$ was converted to a $Cu^{II}$. Then, the obtained $Cu^{II}$ and sulfinic acid anion undergoes a single-electron-transfer (SET) process to produce sulfonyl radical and $Cu^I$. Meanwhile, the enamine intermediate **B** was produced from the tautomerization of copper imine intermediate **A**. The sodium sulfinate forms a sulfonyl radical under the action of $K_2S_2O_8$ and $Ag_2CO_3$ and attacks the intermediate **B** to form the intermediate **C**. Subsequently, an intramolecular SET process caused intermediate **C** to form intermediate **D**. Lastly, the water in the system attacks the carbon cation and the target product **3a** was produced.

**Scheme 5.** Plausible Mechanism.

## 3. Experimental

### 3.1. Materials

Chitosan (≥95% deacetylated) was supplied by the Aladdin Reagent Company (99%, Shanghai, China). Sodium sulfinates were provided by Alfa Aesar (99.5%, Shanghai, China). All chemicals were used as purchased without further purification.

### 3.2. Synthesis of Biomass-Derived Cu Catalysts

Cupric acetate anhydrous (90.83 mg, 0.5 mmol) was dissolved in $H_2O$ (40 mL). Then Chitosan (690 mg) was added to the solution to obtain a suspension that was stirred at 50 °C for 3h. After the mixture was cooled, $H_2O$ was removed slowly under reduced pressure. The obtained light blue colored solid was dried for 12 h at 60 °C under vacuum.

The dried sample was then transferred to a porcelain boat and placed in the oven. The oven was placed under a vacuum and then rinsed with nitrogen for half an hour. The oven was then heated to an appropriate temperature (e.g., 300, 400, or 500 °C), with a temperature gradient of 2 °C/min, and kept at the same temperature for 2 h in a nitrogen atmosphere. After that, the oven was cooled to 25 °C. During the whole process, nitrogen was continuously purged through the oven. The prepared catalyst was stored in a small bottle with a screw cap without any special air protection at room temperature.

### 3.3. Preparation of Ketone Hydrazones

Acetophenone (4 mmol), N-Aminomorpholine (4.4 mmol), 0.5 mL acetic acid, and 10 mL ethanol were added to a reaction tube. Then, the reaction mixture was stirred under reflux for 4 h, and detected by TLC. The mixture was cooled with ice water, and the product was precipitated. The precipitate was filtered and washed three times with petroleum ether and ice water to give the target acetophenone hydrazone product **1**.

### 3.4. Heterogeneous Cu-Catalyzed Sulfonylation of Ketone Hydrazone with Sodium Sulfinates

$Cu_xO_y$@CS-400 (30 mg) was added to the solution of acetophenone hydrazone **1** (0.2 mmol), sodium sulfinates **2** (0.4 mmol), silver carbonate (20 mol%), and potassium persulfate (2.0 equiv) in acetone (1.5 mL), The reaction mixture was stirred at 80 °C for 5 h and detected by TLC. The mixture was filtered, and the solvent was removed under reduced pressure. The gathered residue was purified by column chromatography. Finally, the separated catalyst was washed with ethyl acetate and water, dried for 3 h at 80 °C, and then reused in the next run.

### 3.5. Characterization

Melting points were determined using an X-5 Data microscopic melting point apparatus (Gongyi Corey Instruments Company, Zhengzhou, China). $^1$H NMR and $^{13}$C NMR spectra were recorded using a Bruker Advance 500 spectrometer (Bruker, Saarbrücken, Saarland, Germany) at ambient temperature with CDCl$_3$ as solvent unless otherwise noted and tetramethylsilane (TMS) as the internal standard. IR spectra were recorded using a Nicolet 380 FTIR spectrophotometer (Nicolet, Madison, WI, USA) using KBr discs. Transmission electron microscopy (TEM) images were acquired using a Hitachi HT-7700 microscope (Hitachi, Tokyo, Japan). The X-ray diffraction (XRD) analysis was carried out using a Bruker D8 X-ray diffractometer (Bruker, Saarbrücken, Saarland, Germany) with Cu Ka radiation. Thermogravimetric analyses were performed using a Netzsch STA409PC analyzer (Netzsch, Selb, Bavaria, Germany) at 10 °C/min in air (10 mL/min). A total of 5 mg of each sample in an alumina pan was analyzed in a temperature range of 30–650 °C. Analytical thin layer chromatography (TLC) was performed using Merk precoated TLC (silica gel 60 F254) plates (Merk, Darmstadt, Hessian, Germany).

## 4. Conclusions

A novel copper catalyst has been prepared and characterized from readily available chitosan. This heterogeneous $Cu_xO_y$@CS-400 catalyst showed good reactivity for the selective C(sp$^3$)–H-directed sulfonylation of ketone hydrazone with commercial sodium sulfinates. By using this method, both substituted aryl and alkyl sulfones could be successfully converted to β-ketosulfones. Importantly, the heterogeneous catalyst can be easily recovered and reused at least five times with good activity. Further efforts to understand the heterogeneous C–S coupling mechanism are currently underway in our laboratory.

**Supplementary Materials:** The following supporting information can be downloaded at: https://www.mdpi.com/article/10.3390/catal13040726/s1, Figure S1: Infrared spectra of (a) Chitosan (CS); (b) Cu(OAc)2@CS; (c) CuxOy@CS-300; (d) CuxOy@CS-400; (e) CuxOy@CS-500. Figure S2: Thermogravimetry (TG) of CuxOy@CS-400. Table S1: Control experiments. Table S2: Optimization of Cocatalysts. Table S3: Optimization of time. Scheme S1: Investigation of Influence of Atmosphere on the Reactions. Figures: Characterizations of the Products ($^1$H NMR). References [7,16,17,36,37,43,47,64–74] are cited in the supplementary materials.

**Author Contributions:** Conceptualization, C.S. and Q.Z.; methodology, J.Q. and H.J.; validation, J.Q. and H.J.; formal analysis, K.Z. and Z.L.; investigation, J.Q.; resources, J.Q. and W.Y.; datacuration, C.S. and Q.Z.; writing-original draft preparation, J.Q.; writing-review and editing, J.Q. and Q.Z.; supervision, C.S. and A.J.; project administration, C.S. and Q.Z.; funding acquisition, C.S. All authors have read and agreed to the published version of the manuscript.

**Funding:** This work was supported by the National Natural Science Foundation of China (No. 21302171), the Zhejiang Provincial National Natural Science Foundation of China (No. LQ21E020003), the Zhejiang Shuren University Basic Scientific Research Special Funds (2020XZ011), the "Ten-thousand Talents Plan" of Zhejiang Province (2019R51012).

**Data Availability Statement:** The data are contained within the article.

**Acknowledgments:** We acknowledge Kai Zheng from the College of Biology and Environmental Engineering, Zhejiang Shuren University for his help with SEM, XPS, and FTIR measurements. We also thank Qianfeng Zhang from the Institute of Molecular Engineering and Applied Chemistry, Anhui University of Technology, for his guidance on this work. We also thank Chao Shen from the College of Biology and Environmental Engineering, Key Laboratory of Pollution Exposure and Health Intervention of Zhejiang Province, Zhejiang Shuren University, for his good ideas concerning this work.

**Conflicts of Interest:** The authors declare no conflict of interest.

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
