# Peer review of "Heterogeneous Chitosan@copper Catalyzed Selective C(sp3)–H Sulfonylation of Ketone Hydrazones with Sodium Sulfinates: Direct Access to β-Ketosulfones"

_catalysts, doi:10.3390/catal13040726_

Round 1
Reviewer 1 Report (Previous Reviewer 3)
The authors have correctly answered the various questions about the content of the manuscript, which can now be published in its latest version.
Author Response
We thank the reviewer for the positive comment.
Reviewer 2 Report (Previous Reviewer 4)
The authors appear to have addressed the major concerns of the referee. Some proofreading is still required (e.g., "ketone hydrozone" should be changed to "ketone hydrazone" throughout the manuscript. I also note that references 40 and 66 are the same.
Author Response
Thank you very much for the comment. And then, "ketone hydrozone" has been changed to "ketone hydrazone" throughout the manuscript. Reference 66 has already been removed.
Reviewer 3 Report (New Reviewer)
Zhang and co-workers reports the sulfonylation of ketone hydrazones with sodium sulfinates by using chitosan@copper as catalyst. Author synthesized the catalyst and characterized it. The substrate scope of the reaction is studied well. However, substrate scope was not investigated with sodium sulfinates containing electron withdrawing group such as nitro or cyano. Author performed few control experiments to support the mechanism of reactions. They also performed some derivatization of final sufonyl compound. The importance of work is that author used biomass-derived catalyst for the synthesis of desired compounds. Therefore, would recommend this work for publication after major revision.
1. Author should provide the reason why they got trace compound with CuxOy@CS-500 in optimization table 1.
2. Should write the mole percent of catalyst rather than writing the amount in table 1, 2 and in supporting information.
3. The authors are encouraged to probe whether sulfinate containing electron withdrawing group such as nitro or cyano could be used in the reactions.
4. Authors are encouraged to probe whether bulky group such as alkyl or bromo at ortho position of ketone hydrazone could be used in the reaction.
5. Author should add control experiments with H2O18 in scheme 4 rather than writing it in supporting information.
6. Author did not pick the solvent peak in all 13C spectra. They should process the spectra with solvent peak.
7. In most of 1H spectra peak height is very low and need to be increase.
Author Response
We thank the reviewer for the positive comment. The manuscript has been carefully checked and revised.
Comment 1: Author should provide the reason why they got trace compound with CuxOy@CS-500 in optimization table 1.
Response: Thank you very much for the comment. Due to the calcination temperature of CuxOy@CS-500 has been changed, maybe, the content of the active components in the catalyst has been changed.
Comment 2: Should write the mole percent of catalyst rather than writing the amount in table 1, 2 and in supporting information.
Response: Thanks to the reviewer for comment. In our opinion, the mole percent of Cu catalyst was not very rigorous. Because by multifaceted characterization of the as-prepared catalysts, we know that the existence of Cu, Cu2O and CuO in the catalyst. In Table 1, indicates that Cu2O, which really help catalyze sulfonylation reaction. Hence, further efforts to understand and control the Cu2O contents on catalysts in our laboratory.
Comment 3: The authors are encouraged to probe whether sulfinate containing electron withdrawing group such as nitro or cyano could be used in the reactions.
Response: The sulfinate containing electron withdrawing group such as nitro or cyano has been used in the reactions and trace of the corresponding products was formed.
Comment 4: Authors are encouraged to probe whether bulky group such as alkyl or bromo at ortho position of ketone hydrazone could be used in the reaction.
Response: Thank you very much for your comment. When using the ortho- ubstituted ketone hydrazone (F) as the substrate, the corresponding product were formed in 70% yields. when the ketone hydrazone bearing substituents at meta position (Cl and CH3), the products 3c and 3d were furnished in good yields of 77% and 81% yields. We will further efforts to explore the reaction influence of the bulky group such as alkyl or bromo at ortho position of ketone hydrazone.
Comment 5: Author should add control experiments with H2O18 in scheme 4 rather than writing it in supporting information.
Response: Thank you very much for your comment. The control experiments with H2O18 has been added to the manuscript.
Comment 6: Author did not pick the solvent peak in all 13C spectra. They should process the spectra with solvent peak.
Response: The solvent peak in all 13C spectra have been picked.
Comment 7: In most of 1H spectra peak height is very low and need to be increase.
Response: Thank you very much for your comment. All the keto sulfone products are known compounds and the result of the NMR test match many literature.
Round 2
Reviewer 3 Report (New Reviewer)
The authors have addressed all the comment. Manuscript has been sufficiently improved to publish in Catalysts.
This manuscript is a resubmission of an earlier submission. The following is a list of the peer review reports and author responses from that submission.
Round 1
Reviewer 1 Report
This paper describes the preparation of Cu-chitosan composite catalyst and its application for the catalytic alpha-sulfonylation of ketone hydrazones. The same catalytic oxidative sulfonylation reaction of ketone hydrazine using copper salt was reported in 2021. Although chitosan-supported copper complex catalysts have been continuously studied by the authors, the pyrolyzed chitosan-Cu catalyst is newly prepared, so this manuscript can be published as a new entry of sustainable heterogeneous catalyst. However, the submitted manuscript lacks many important information to judge whether the reported results are meaningful or not, and contains many incorrect explanations which causes severe confusion for readers. Thus, the reviewer does not recommend to the editor to accept this manuscript at this stage and wish the authors to revise the manuscript carefully.
Additional minor comments are listed below,
Comments:
1. The reaction substrate used in this manuscript is “(ketone) hydrozone”, not “ketoxime.” This mistake was seen in many places in the manuscript including the title. At the latter part of the manuscript, “methoxy oxime” also appeared, which would also be hydrazone.
2. At the line 17 in page 1, “functionalized aryl and alkyl were converted” is confusing. For example, “aryl and alkyl sulfinates were applicable to the reaction” is better to understand. At least, please use “sulfinate”, not only “aryl and alkyl”.
3. At the line 33 in page 1, “sodium sulfonates” should be “sodium sulfinate”.
4. In Scheme 1c, the authors show the equation of ref37, but ref 23 and 24 are more related and ref 52 is almost same reaction to the present catalytic reaction. So, this equation and related sentences should be replaced to indicate the background more clearly.
5. At the last sentence in page 1, the authors should refer other heterogeneous catalysts for this type reactions. For example, Appl. Organomet. Chem. 2019, 33, e5001.
6. At the line 89 in page 3, the vibration band at 1658 cm-1 is widely considered as the amide vibration of remaining acetyl groups, as can be seen other reported papers. The authors need to assign more carefully based on the reported results.
7. In Figure 2a-d, color of scale bars and scales should be changed to more visible color.
8. In Figure 3, text in figures are too small to read. Please enlarge them for better clarity.
9. For Figure 3e, the authors need to explain what is concluded from XRD. There is no explanation for XRD pattern. It is meaningless.
10. From the line 123 to 126 in page 5 and entries 1-3 in Table 1, there is inconsistency between CuxOy@CS-300 and -400. Maybe Table 1 is correct.
11. In Table 1 and related sentences, there is no information about the catalyst loading. This is indispensable to discuss the catalytic activity. The catalyst loading of CuxOy@CS-300, -400, and -500 must be different when 30 mg of these catalysts are used, as can be clarified by the TG data. Cu atom % of each catalyst must be calculated based on the TD data and shown in Table 1.
12. Table 1 and 2 lack table footnotes explaining detail information for reaction conditions, although there are footnote signs, such as a, b, c, and d. In case of Table 2, the latter part of the table footnote is disappeared.
13. Related to Table 1, there is no explanation about the necessity of Ag2CO3. What happen without Ag2CO3 or other Ag salt? This is very important to discuss the reaction mechanism.
14. At lines 166 and 168 in page 7, 3m-3q and 3r-3v are opposite.
15. In Scheme 3, what is the meaning of ref38 at the first step? Ref 38 does not describe this type of reaction.
16. At the bottom equation of Scheme 3 showing large scale reaction, is the catalyst amount of 30 mg correct? Generally, when we scale up reactions, the amount of catalyst is also scaled up. If 30 mg is correct, this result shows that the catalyst shows much higher turnover number for the reaction.
17. At lines from 184 to 188 in page 8, hot filtration experiment is described. But, the described information is not enough to judge catalyst leaching happens or not. If the reaction was completed after 2 h, the hot filtration is meaningless. To show the result more clearly the authors would better to show the time-course reaction profile both of standard condition and with hot filtration at the time point of almost half conversion.
18. At the line 195 in page 9, “radical collecting reagent” should be “radical trapping reagent”.
19. For the proposed mechanism, there many issues to be addressed. Molecular structure of D in Scheme 5 is different from D in Scheme 4. Which is correct? If enamine structure is correct, how enamine formed from the intermediate C? Please add the discussion.
The authors describes the generation of Cu(II) species by the single electron transfer process, but Cu(III) is drawn for the proposed mechanism. Which is correct?
In Table 1, no ketosulfone was formed without copper catalyst. In this case, how about the recovery of hydrazone and is there by-product? I’m wondering the generation of enamine without copper catalyst and radical sulfonylation takes place to afford the alpha-sulfonylated hydrazine.
20. At the line 213 in page 10, what is O-methylhydroxylamine? It cannot be found in the proposed mechanism.
21. Incorrect English grammar is observed at several places. The authors need to check the English carefully or ask English correction before submission.
Reviewer 2 Report
This manuscript by Zhang and co-workers describes a new useful heterogeneous catalyst for ketoxime a-sulfonylation. The heterogeneous chitosan-based catalyst including copper reaction site is revealed to promote the reaction in efficient manner, and the target products can be obtained by easy separation of the catalyst and simple workup. The catalyst thus recovered can be repeatedly used for the same reactions without significant loss of the catalytic activity. In this work, the catalyst structure is detailly discussed from the obtained data by spectroscopic analysis and microscopic techniques. The synthetic scope and advantages are suitably demonstrated, which are in accord with required standard of synthetic work. I recommend provisional acceptance of this work in Catalysts journal.
The following points need to be addressed;
1. The manuscript lacks appropriate mechanistic information, and the catalytic cycle and discussion suggested by the authors have several unacceptable points. The authors’ mechanism is quite different from those suggested for homogeneous catalyst in the same reaction (see early study in ref. 52). The mechanistic difference should be rationalized with suitable discussion and experimental proof.
2. The same transformation using catalytic copper(II) oxide with silver carbonate was reported earlier. Therefore, the introductory part discussing previous works should include this information. Somehow, this early study is missing from Scheme 1.
3. Regarding Scheme 3, reference 38 does not deal with hydroxylamine-mediated transformation. The unique aromatization reaction applied to 4a and two-step quinoxaline synthetic procedure leading to 5b were reported by others (Chang et al. Org. Lett. 2015, 17, 3142; Org. Lett. 2019, 21, 1832.).
4. Regarding the compatibility of chitosan-based catalyst, availability of the substances with acidic or basic functionality seems to be an interesting option for extending the substrate scope.
5. The footnote a in Table 1 is somewhat missing from the contents.
Reviewer 3 Report
The manuscript by Qianfeng Zhang describes a copper-based catalyst adsorbed on chitosan and its application for CH activation of ketoximes with arylsulfinates. The catalyst is well characterised and the study of the reaction is well described.
The manuscript can be published in the journal Catalysts if the following points are considered.
A major revision of the English language and style is required. For example, see a sentence beginning with "And" in the abstract.
On page 2, line 61, the authors say that "(their catalyst) encounters several challenging issues, such as activity, stability and selectivity for various organic reactions". Explain this concept better with examples.
The reaction works better in acetone than in other solvents. Have the authors made a hypothesis?
The authors also give no explanation as to why the aliphatic sulfinates show poor reactivity.
A catalyst that has to be prepared by calcination at 400 °C and used in an acetone solution in the presence of 20% Ag2CO3 and 2 equivalents of K2S2O8 cannot be called "sustainable". It is a nice heterogeneous catalyst for an interesting reaction, but please delete the word "sustainable" from the manuscript.
Reviewer 4 Report
The authors have modified a reported procedure for achieving sulfonylation of acetophenone N-morpholinohydrazones to give beta-keto sulfone products. This work appears to be directly derived from the procedures given in reference 52 except that in this submission the authors have prepared a supported form of the copper catalyst by pyrolyzing Cu(OAc)2 with chitosan. The resulting chitosan supported Cu provided a solid-state source of Cu catalyst for the sulfonylation reaction that could be recovered and reused. Notably, the reaction also requires the use of a Ag co-catalysts and excess potassium persulfate. Given the relative cost of Cu and Ag, the "green chemistry" value of Cu@chitosan catalysts is somewhat questionable. Moreover, the work really is essentially a direct reproduction of the chemistry appearing in reference 52 except that a chitosan-supported Cu catalyst is used in place of discrete copper complexes. The only novel aspects of this manuscript, then, are the preparation and characterization of the Cu@chitosan assemblies along with demonstration of efficacy in a known reaction. As such, this work appears to constitute a minimally incremental advance in the area of catalysis. Several other issues to consider:
1. Paper needs to be carefully proofread - there are confusing statements that can obscure meaning. For example, lines 140-144; 163-164; amd top of page 9. This last section is especially confusing as the authors mention reactivity of oximes, but no oximes were used in this study (the substrates are N-morpohlinohydrazones - presumably because that's what the authors in reference 52 used).
2. I note that the control studies reported in Scheme 4 are exactly the same control studies performed by the authors of reference 52.
3. Likewise, the plausible mechanism illustrated in Scheme 5 is clearly based on the mechanism advanced in reference 52 except it is unclear why a two-electron oxidative addition of copper into the N-N bond is proposed rather than a SET (which seems consistent with typical chemistry of Cu). Additionally, the authors indicate that enamine D is formed under their reaction conditions (Scheme 4), but fail to show this intermediate in the catalytic cycle. Instead, the authors illustrate a highly unlikely reductive elimination to reform the N-N bond cleaved in the first step of the cycle.
4. The authors state that 10 mol% catalyst was used in their reactions. What is this based on - Cu content of the catalyst? How was this determined?
5. All the keto sulfone products are known compounds - references to their prior preparation should be provided in the SI.
6. In reporting HRMS the calculated formula should include the "extra" proton from positive ion ESI.